# Prevalence of Antibiotics Prescription and Assessment of Prescribed Daily Dose in Outpatients from Mexico City

**DOI:** 10.3390/antibiotics9010038

**Published:** 2020-01-20

**Authors:** Ramiro Sánchez-Huesca, Abel Lerma, Rebeca M.E Guzmán-Saldaña, Claudia Lerma

**Affiliations:** 1CEBECI Farmacología Clínica Sociedad Civil, 06700 Mexico City, Mexico or; 2Institute of Health Sciences, Universidad Autónoma del Estado de Hidalgo, 42160 San Juan Tilcuautla, Mexico; abel_lerma@uaeh.edu.mx (A.L.); remar64@yahoo.com.mx or; 3Department of Electromechanical Instrumentation, Instituto Nacional de Cardiología Ignacio Chávez, 14080 Mexico City, Mexico

**Keywords:** antibiotics prescription, pharmacoepidemiology, outpatients, prescribed daily dose, defined daily dose, drug utilization

## Abstract

Pharmacoepidemiological research about antibiotics is supported by the World Health Organization (WHO), but data regarding antibiotic prevalence based on actual prescriptions and dosing patterns are insufficient. The aims were: (i) To estimate the prevalence and prescribed daily dose (PDD) of antibiotics in outpatients from Mexico City and (ii) to compare the PDD against the defined daily dose (DDD), as established by the WHO. The study included 685 prescriptions of antibiotics selected randomly from five geographical zones of Mexico City. Drug, dose, frequency, and duration of treatment were obtained from each prescription. PDD values of each antibiotic drug were calculated as the average of the daily doses. Sub-use and overuse were determined by the ratio PDD/DDD for each prescription. The most prescribed antibiotics to outpatients from Mexico City included six pharmacological groups: quinolones (28%), penicillins (23%), cephalosporins (17%), macrolides (10%), lincosamides (9%), and sulfonamides (4%). Both overuse and sub-use were high (55% and 63%, respectively). In conclusion, most of the antibiotics with a high prevalence of prescription also had a high rate of either sub-use or overuse, with prescribed doses that significantly differ with their corresponding DDD. The dosing variation has important clinical implications since it denotes low prescription control.

## 1. Introduction

Antibiotic therapy has marked a threshold in modern medicine and the way infectious diseases are treated, but these drugs have represented a challenging issue since the beginning of their use in big populations, and therefore, global policies have taken place to reduce the actions that lead to an incorrect use [1,2]. As an example of global policies, the World Health Organization (WHO) has supported the antimicrobial stewardship since 1990, as an effort to improve and maintain an adequate utilization [3,4], since it is well recognized that misuse may cause bacterial resistance [5,6,7], serious adverse effects [8], increase mortality rates [9,10], and have a negative effect on the economy for both patients and health services [11,12]. 

It seems that the best way to enclose antibiotic-related problems is to have a strict drug control [1,6,13], and it is in this matter that huge differences are found among countries [11], and even in a particular geographical area [14,15], so the problem has not been contained. Antibiotic control should be made in two different setups: prescription and dispensation [16,17]. Particularly, Mexico restricted dispensation in 2010, when antibiotic retailing changed [13]. Before that year, antibiotics could be retrieved as if they were over-the-counter (OTC) medications [2,18], but since the law update, it is mandatory to present a prescription to the pharmacy for the antibiotic to be dispensed. 

However, there is not a registry of any control in prescription, so there is still a long way to go until we could achieve good practices related to antibiotic use. In fact, there is poor evidence of antibiotic utilization in Mexico [1], and most of the research is based on retail sales [18,19], but not in actual prescriptions, which could be used to relate antibiotic utilization in the human clinical setting and the doses to estimate the prevalence. Thus, we have not been able to characterize the whole problem regarding antibiotic utilization in Mexico. Furthermore, prevalence rates based on prescriptions are expected to be useful to match those rates with data that already shows bacterial resistance in multiple strains [20]. This study aimed to describe the antibiotic use by the estimation of prevalence in an outpatient setting and identify a current discrepancy between two pharmacoepidemiological indicators: prescribed daily dose (PDD) and defined daily dose (DDD), as reported by the WHO.

## 2. Results

### 2.1. Prescription Prevalence

Antibiotic prescription is widespread in Mexico City and includes the utilization of 12 pharmacological groups (Table 1). The most prescribed antibiotics were quinolones, followed by penicillins, cephalosporins, macrolides, lincosamides, and sulfonamides. In most of the pharmacological groups, approximately 75% of prescriptions correspond to 1 or 2 drugs only. For example, the most prescribed quinolones were ciprofloxacin (45%) and levofloxacin (34%), while the most prescribed penicillins were amoxicillin (38%) and its combination with a beta-lactamase inhibitor clavulanic acid (36%). These results indicate that, in spite of the great variability in the prescription of antibiotics and pharmacological groups, there was a narrow diversity of drugs within each group.

### 2.2. Assessment of Prescribed Daily Dose

Table 2 shows the PDD of the 27 prescribed drugs in our sample, which are compared with the DDD, as reported by the WHO. There was a statistical difference between PDD and DDD in 14 drugs (levofloxacin, nalidixic acid, amoxicillin whether it is alone or in combination with a clavulanic acid, dicloxacillin, cefalexin, ceftriaxone, cefuroxime, azithromycin, clarithromycin, erythromycin, clindamycin, lincomycin, and gentamicin) of 6 pharmacological groups (quinolones, penicillins, cephalosporins, macrolides, lincosamides, and aminoglycosides). 

The PDD divided by the DDD was calculated to determine the discrepancy of dose utilization from a statistical point of view (Table 3). We considered sub-use or overuse as variables according to the difference from the unit: when the prescribed dose of a particular antibiotic matches with the one reported by the WHO, then the quotient is 1.0, but if the prescribed dose is less than 1.0, then we say that there is sub-use or overuse if the value is greater than 1.0. Overuse occurred in 15 out of 27 prescribed antibiotics, and this represents 55% of all anatomical, therapeutical, and chemical (ATC) classes. For the doses, the drugs that showed the greatest level of overuse are amoxicillin, either alone or in combination with clavulanic acid, azithromycin, levofloxacin, and clarithromycin. The sub-use was present in 17 out of 27 drugs, which is equivalent to 63% of all ATC classes, with the greatest sub-used antibiotics being clindamycin, ceftriaxone, cefalexin, ampicillin, and dicloxacillin.

Table 4 shows that beta-lactams (penicillins and cephalosporins) were the drugs with either the greatest sub-use or overuse and a frequency of 234 (34%) prescriptions, followed by macrolides whose variations represent 94% as sub-use or overuse. Moreover, huge discrepancy rates were presented regarding both lincosamides and quinolones when comparing PDD and DDD values.

Prescription discrepancies related to doses are also described by their normalized values of PDD with respect to DDD (Table 5). The normalized PDD for each drug means that if the value is zero, then PDD and DDD are mathematically equal; therefore, contrasting zero with the median of the normalized dose infers a difference within doses. There were 13 antibiotics with significant discrepancies (*p* < 0.05). Some antibiotics had large discrepancies (i.e., median normalized PDD values farther away from zero), including cefuroxime, clarithromycin, amoxicillin plus clavulanic acid, azithromycin, and lincomycin, while other antibiotics had small discrepancies (i.e., median normalized PDD values closer to zero), including levofloxacin, cefalexin, clindamycin, gentamicin, and fosfomycin.

## 3. Discussion

There are only very few studies that assess the antibiotic prescription in Mexico [1,20]. For example, one investigation based on retail sales showed an increased rate in antibiotic consumption between 2007 and 2012 [18], especially since 2010, when the dispensation was restricted only to patients who presented a prescription [13]. Another study about dispensation of antibiotics in a city in the border between Mexico and the United States of America demonstrated that antibiotics were the pharmacological group with the highest level of sales (65% without a prescription), and this fact raised a concern about their inappropriate utilization, which in several cases was based on advice from pharmacy clerks whose educational level is usually low [21,22]. Other studies explore antibiotic prescription and its appropriateness in clinical settings of outpatients [23,24], or the antibiotic resistance to pathogen bacteria in urinary tract infections [25,26]. However, this study is the first research in Mexico that focuses on the prevalence of prescription and assessment of the prescribed doses of antibiotics, which is different from assessing the consumption of antibiotics or bacterial resistance.

In this study, quinolones represented the pharmacological antibiotic group with the greatest prescription prevalence in Mexico City. This fact differs from other countries where penicillins are the most prescribed pharmacological group, as it is the case in Egypt [27], although it seems to be a phenomenon in developing countries. Typically, the most economically developed countries have a higher consumption of antibiotics [19], which has changed the prevalence tendencies in time, so these countries are more likely to increase the utilization of pharmacological groups with a wider antibiotic spectrum (such as cephalosporins and quinolones) [14], as it happens in the United Kingdom, Germany, or Spain [15]. Nevertheless, there is relevant evidence that places Mexico as a country without a high antibiotic consumption in comparison to other countries [19], even when our results clearly show that there is a pharmacoepidemiological tendency of antibiotic utilization that corresponds to a country of higher economic development.

The use of quinolones as the most prescribed drugs gives an interesting and overt pattern to future studies to relate the diagnosis with the pharmacoepidemiological results of prevalence in the outpatient setting, since it is necessary to further analyze the consequences of prescribing a given antibiotic for a determined infection. Pharmacoepidemiology establishes a specific methodology for this kind of study called prescription–indication [16,28,29], which is extremely useful in the daily clinical practice since they show the adequacy of prescription. At the same time, concerning antibiotics, they are also important to foresee adverse effects and bacterial resistance [5,6].

The variability of prescription among the different pharmacological groups in this study was shown to be relatively low, as 7 out of 10 prescribed antibiotics in Mexico City belong to either a quinolone or beta-lactams (penicillins and cephalosporins), which is certainly a highlight when comparing our results to other regions around the world where diversity in prescription is wider [15]. However, the prescription of five pharmacological groups have been identified as high-prevalence antibiotics (penicillins, macrolides, quinolones, cephalosporins, and sulfonamides) [14,15].

The assessment of antibiotic utilization and the presence of problems related to them are difficult aspects to quantify. Nevertheless, the research in this subject should consider factors that involve both the prescriber and the patient and, in this sense, the differences that may be observed in antibiotic dosing after the statistical comparison of the PDD and the theoretical DDD might explain the point of view of the factors mentioned herein. For example, among general practitioners, several factors are associated with the antibiotic prescribing volume, including appointment duration, training practice, as well as the prescriber’s age and sex [30]. On the other hand, the factors that are directly related to the patient are key to fully understand prescriptions’ rationale and patterns, such as the actual antibiotic consumption, which can be expressed as therapeutic adherence or compliance to directions from health providers [28,31]. In summary, antibiotic utilization is complex, but it is probably the first needed approach to understand the distribution in terms of prevalence and doses, and that is why this study contributes to the assessment of antibiotic utilization expressed throughout these two variables.

Indeed, drug utilization has become a crucial matter regarding the assessment of antibiotics, especially when doses are considered into the evaluation, since it allows us to take adequate strategies into account to encourage health promotion to decrease and avoid antibiotic-related problems. Particularly, bacterial resistance has been identified as a globally recognized problem in public health related to the incorrect use of antibiotics [25,26,32]. Bacterial resistance occurs in a higher proportion when they are administered in inadequate doses (e.g., when posology was prescribed in doses below the therapeutic ranges or during periods too short) [7,33]. As mentioned above, these two factors regarding the posology of prescription are dependent on both the prescriber and the patient, so this study indirectly assesses the prescriber by the comparison of the PDD values against their corresponding DDD. Therefore, it is especially interesting to assess those drugs that showed sub-use, in this case, the 63% of all ATC classes, particularly clindamycin, ceftriaxone, cefalexin, ampicillin, and dicloxacillin, and at the same time, we can leave the hypothesis stating that those antibiotics that showed overuse may represent a higher risk for the presence of adverse effects.

The prescriptions did not include the diagnosis since it is not a requirement in the Mexican regulation and because of that, it was not possible to identify further reasons about the discrepancies between PDD and DDD, for example, if an antibiotic was prescribed for an indication different to the main one or if the patient was being treated concomitantly with other medications or was in a medical condition that oriented the physician to reduce the antibiotic daily dose [34].

Antibiotic use, including prescription and administration, must reach adequacy as a key factor to avoid bacterial resistance. However, the correct utilization represents itself as a complex problem that involves the right antibiotic selection, proper posology, therapeutic adherence, and patient compliance. There is an urgent need in Mexico to continue researching about antibiotics to create a solid reference regarding the pharmacoepidemiology of antibiotics that aid future health decision-making processes [6]. Thus, studies of this kind should be replicated in larger samples and in different settings, such as inpatient populations, but also in less-studied samples, as is the case of pediatric or immunosuppressed patients.

This study is part of the initial efforts in giving an overview and solution to antibiotic utilization in Mexico, focused on the outpatient population through the application of a formal pharmacoepidemiological methodology to get robust outcomes about drug utilization patterns and foresee better prescription practices that decrease adverse effects, inefficacy, and strongly avoid bacterial resistance. Nonetheless, considering the massive consumption of antibiotics, future studies are highly needed, especially those that explore the prevalence of antibiotics based on prescriptions in a multicentric fashion, since this ensures that the medication was actually dispensed to the patient, not in retailing sales, but also to encourage the research that explores the relationship between the antibiotic spectrum with the prescription (for example, utilization of penicillin for the treatment of gram-positive bacteria) [20], the prescriber and patient characteristics, and the associated factors to prescription of inadequate doses.

## 4. Materials and Methods 

### 4.1. Study Protocol

This study was retrospective, cross-sectional, descriptive, and observational, with a sample based on prescriptions from outpatients, and all data were retrieved from community pharmacies. There are not available databases in Mexico. Therefore, for creating the first database, we applied a sampling technique creating clusters to be able to sample data from a huge geographic zone, such as Mexico City with more than 8 million inhabitants, this technique is also useful to avoid bias since all data are randomly obtained with the same probability to be chosen and considers the population density at the moment of data collection, so representativity is guaranteed. 

For this study, Mexico City was divided into five geographical zones (north, south, east, west, and center), then, 25 community pharmacies along the city were selected from a predetermined list of pharmacies using a table of random numbers (i.e., five pharmacies for each zone). We proceeded to data collection once the pharmacies were selected, prescriptions being our unit of analysis, it is important to mention that there is not an electronic record of prescriptions in Mexico so the pharmacy must store the prescription in a hard copy (printed on paper). Previously, we calculated a sample size according to the proportions formula and considering a 20% loss in case of illegibility of prescriptions, those with incomplete posology data or prescriptions different from antibiotics or pediatric patients. Thus, the calculus was a total of 685 prescriptions which were divided equally in the 25 community pharmacies (77 prescriptions per pharmacy).

Regarding the inclusion criteria, they were prescriptions delivered by physicians or odontologists, from adult patients (older than 18 years old) and with at least one antibiotic. All prescriptions were selected using random numbers generated by a computer. In this way, we could ensure that all obtained data were from outpatients from Mexico City.

The protocol of the present study complies with the national and international ethical aspects as well as applicable confidentiality laws in Mexico. The Committee of Ethics for Research of the Medical Research Center CEBECI Farmacología Clínica, Sociedad Civil approved the study (protocol number ICE-1506-NIF).

### 4.2. Prescription Analysis

We extracted the whole posology from the prescription, considering the drug, the quantity of drug per dosage form, frequency of administration, and duration of treatment as independent variables. Afterward, the drugs were classified according to the anatomical, therapeutical, and chemical (ATC) coding, as indicated by the WHO, and each code has been linked with its corresponding value of DDD published on the WHO website in 2018 [35].

All DDD values were present in the WHO database except for the combination of sulphamethoxazole and trimethoprim, for which the value used as DDD was the one of maximum daily dose, as published in the information to prescribe (ITP), this is in the technical document from the main pharmaceutical laboratory that manufactures the medication containing this drug combination [36]. 

PDD was calculated as the product of the antibiotic dose per dosage form by the number of dosage forms indicated in each administration by the frequency in a day. This calculus was made for each one of the prescriptions of the same drug, and then the corresponding PDD mean was obtained and considered for the general statistical analysis.

### 4.3. Statistical Analysis

The antibiotic prevalence was calculated through the division of the number of prescriptions by the total number of prescriptions. The same analysis was made for the diverse pharmacological groups, and the results are presented as prevalence values and 95% confidence interval. According to Kolmogorov–Smirnov tests, most data did not have a normal distribution, and the results are presented as the median (percentile 25–percentile 75).

The pharmacoepidemiological assessment of prescribed doses of antibiotics was done by comparing the values of medians of PDD for each ATC class against its corresponding DDD using the Wilcoxon signed-rank test. The ratio between PDD and DDD was also calculated for each prescription, to determine sub-use (defined as PDD/DDD < 1.0) or overuse (defined as PDD/DDD > 1.0) for all prescribed antibiotics. 

The doses were also normalized taking the PDD minus the DDD as the numerator and the DDD being the formula (PDD-DDD)/DDD and considering that there is no discrepancy between PDD and DDD if the value is zero as the denominator. The Wilcoxon signed-rank test was also performed comparing a value of zero against the median of the normalized value for each ATC class. The statistical analysis was performed using SPSS version 21.0 and a value of *p* < 0.05 was considered as significant.

## 5. Conclusions

The analyzed prescription in this pioneering study in Mexico shows that most of the antibiotics with a high prevalence of prescription also had an elevated rate of either sub-use or overuse. The statistical value of the DDD reported by WHO when comparing to PDD does not necessarily reflect the quality of prescription, since the dose may vary according to the diagnosis and particular characteristics from the patient [37,38], but it represents a pattern of antibiotic prescription in Mexico City that shed light to understand the behavior of prescription. The differences found in medians of PDD and DDD are relevant for the evaluation of prescription, especially the fact that there is a statistical difference in at least one of the most prescribed drugs for each pharmacological group. An antibiotic discrepancy was shown in both statistical comparisons of quotients and normalized PDDs. The latter has important clinical implications as it denotes low control in prescription.

The present study shows the utilization of antibiotics in an outpatient setting by the analysis of doses. Mexico does not have any record about this type of data, which is useful to prescribers for whom the knowledge in pharmacoepidemiological utilization tendencies may be a coadjutant daily clinical decision-making process related to prescription. This study and some others of the same class might justify to the Mexican health authorities the application of public strategies towards both health providers and patients to increase the rational use of antibiotics, and in this way, to contribute to the efforts that are globally taking place in other regions.

## Figures and Tables

**Table 1 antibiotics-09-00038-t001:** Prescription of antibiotics in 685 outpatients from Mexico City.

ATC Class–Name	N	Prevalence (95%CI)
J01MA02–Ciprofloxacin	85	45 (38–52)
J01MA12–Levofloxacin	65	34 (14–55)
J01MA14–Moxifloxacin	19	10 (6–14)
J01MA01–Ofloxacin	8	4 (1–7)
J01MB02–Nalidixic acid	7	4 (1–6)
J01MA06–Norfloxacin	6	3 (1–6)
**Total prescriptions of quinolones**	190	28 (24–31)
J01CR02–Amoxicillin plus clavulanic acid	61	38 (31–46)
J01CA04–Amoxicillin	57	36 (28–43)
J01CA01–Ampicillin	20	13 (7–18)
J01CF01–Dicloxacillin	17	11 (6–15)
J01CE09–Procaine benzylpenicillin	3	2 (0–4)
J01CE08–Benzathine benzylpenicillin	2	1 (0–3)
**Total prescriptions of penicillins**	160	23 (20–27)
J01DB01–Cefalexin	47	39 (31–48)
J01DD04–Ceftriaxone	33	28 (20–36)
J01DD08–Cefixime	17	14 (8–21)
J01DC02–Cefuroxime	12	10 (5–15)
J01DD14–Ceftibuten	5	4 (1–8)
J01DD13–Cefpodoxime	3	3 (0–5)
J01DD15–Cefdinir	1	1 (0–2)
J01DD01–Cefotaxime	1	1 (0–2)
**Total prescriptions of cephalosporins**	119	17 (15–20)
J01FA10–Azithromycin	41	58 (46–69)
J01FA09–Clarithromycin	19	27 (16–37)
J01FA01–Erythromycin	6	8 (2–15)
J01FA02–Spiramycin	5	7 (1–13)
**Total de prescriptions of macrolides**	71	10 (8–13)
J01FF01–Clindamycin	50	83 (74–93)
J01FF02–Lincomycin	10	17 (7–26)
**Total prescriptions of lincosamides**	60	9 (7–11)
J01EE01–Sulfamethoxazole plus trimethoprim	26	100 (100–100)
**Total prescriptions of sulfonamides**	26	4 (2–5)
J01XE01–Nitrofurantoin	14	24 (13–35)
A07AX03–Nifuroxazide	2	3 (0–8)
**Total prescriptions of nitrofurans**	16	2 (1–3)
A07AA11–Rifaximin	12	20 (10–31)
**Total prescriptions of rifamycins**	12	2 (1–3)
J01XX01–Fosfomycin	11	19 (9–29)
**Total prescriptions of phosphonates**	11	2 (1–2)
J01AA02–Doxycycline	5	8 (1–16)
J01AA08–Minocycline	3	5 (0–11)
J01AA04–Lymecycline	1	2 (0–5)
J01AA06–Oxytetracycline	1	2 (0–5)
**Total prescriptions of tetracyclines**	10	1 (1–2)
J01GB03–Gentamicin	6	10 (2–18)
J01GB06–Amikacin	2	3 (0–8)
J01GB05–Neomycin	1	2 (0–5)
**Total prescriptions of aminoglycosides**	9	1 (0–2)
J01DH03–Ertapenem	1	2 (0–5)
**Total prescriptions of carbapenems**	1	0 (0–0)

ATC = anatomical, therapeutical, and chemical, CI = confidence interval.

**Table 2 antibiotics-09-00038-t002:** Prescribed daily dose (PDD) and defined daily dose (DDD) for antibiotics with at least five prescriptions. PPD is described as median (percentile 25–percentile 75).

Group	ATC Class–Name	DDD (mg)	PDD (mg)
Quinolones	J01MA02–Ciprofloxacin	1000	1000 (1000–1000)
	J01MA12–Levofloxacin	500	500 (500–750) **
	J01MA14–Moxifloxacin	400	400 (400–400)
	J01MA01–Ofloxacin	400	400 (400–400)
	J01MB02–Nalidixic acid	4000	1500 (1500–1500) *
	J01MA06–Norfloxacin	800	800 (800–800)
Penicillins	J01CR02–Amoxicillin plus clavulanic acid	1000	1750 (1750–1750) **
	J01CA04–Amoxicillin	1000	1500 (1500–1500) **
	J01CA01–Ampicillin	2000	1750 (1750–2000)
	J01CF01–Dicloxacillin	2000	1500 (1500–2000) **
Cephalosporins	J01DB01–Cefalexin	2000	1500 (1500–1500) **
	J01DD04–Ceftriaxone	2000	1000 (1000–1000) **
	J01DD08–Cefixime	400	400 (400–400)
	J01DC02–Cefuroxime	500	1000 (1000–1000) **
	J01DD14–Ceftibuten	400	400 (400–400)
Macrolides	J01FA10–Azithromycin	300	500 (500–500) **
	J01FA09–Clarithromycin	500	1000 (1000–1000) **
	J01FA01–Erythromycin	1000	1500 (1500–2000) **
	JP1FA02–Spiramycin	3000	2250 (2250–3000)
Lincosamides	J01FF01–Clindamycin	1200	900 (900–900) **
	J01FF02–Lincomycin	1800	600 (600–600) **
Sulfonamides	J01EE01–Sulfamethoxazole plus trimethoprim	1600	1600 (1600–1600)
Nitrofurans	J01XE01–Nitrofurantoin	200	300 (200–400)
Rifamycins	A07AA11–Rifaximin	600	700 (600–800)
Phosphonates	J01XX01–Fosfomycin	3000	3000 (1500–3000)
Aminoglycosides	J01GB03–Gentamicin	240	160 (160–160) *
Tetracyclines	J01AA02–Doxycycline	2000	1500 (1500–2000)

** *p* < 0.01 versus DDD, * *p* < 0.05 versus DDD. ATC = anatomical, therapeutical, and chemical.

**Table 3 antibiotics-09-00038-t003:** Relationship PDD/DDD, sub-use (PDD/DDD < 1.0) and overuse (PDD/DDD > 1.0) for antibiotics with at least 5 prescriptions. PDD/DDD quotient is described as median (percentile 25–percentile 75); Sub-use, overuse, and total are described as n (%).

ATC Class–Name	PDD/DDD	Sub-Use	Overuse	Total
J01FA10–Azithromycin	1.67 (1.67–1.67)	0 (0)	41 (100)	41 (100)
J01DD04–Ceftriaxone	0.50 (0.50–0.50)	33 (100)	0 (0)	33 (100)
J01FF02–Lincomycin	0.33 (0.33–0.33)	10 (100)	0 (0)	10 (100)
J01MB02–Nalidixic acid	0.38 (0.38–0.38)	7 (100)	0 (0)	7 (100)
J01FA01–Erythromycin	1.50 (1.50–2.00)	1 (17)	5 (83)	6 (100)
J01GB03–Gentamicin	0.67 (0.67–0.67)	6 (100)	0 (0)	6 (100)
J01CA04–Amoxicillin	1.50 (1.50–1.50)	0 (0)	56 (98)	56 (98)
J01CR02–Amoxicillin plus clavulanic acid	1.50 (1.50–1.50)	1 (2)	57 (93)	58 (95)
J01FF01–Clindamycin	0.75 (0.75–0.75)	45 (90)	0 (0)	45 (90)
J01DB01–Cefalexin	0.75 (0.75–0.75)	36 (77)	4 (9)	40 (86)
J01XE01–Nitrofurantoin	1.50 (1.00–2.00)	3 (21)	9 (64)	12 (85)
J01FA09–Clarithromycin	2.00 (2.00–2.00)	0 (0)	16 (84)	16 (84)
J01DC02–Cefuroxime	2.00 (2.00–2.00)	0 (0)	10 (83)	10 (83)
J01CA01–Ampicillin	0.75 (0.75–1.00)	13 (65)	3 (15)	16 (80)
J01FA02–Spiramycin	0.75 (0.75–1.00)	3 (60)	1 (20)	4 (80)
J01CF01–Dicloxacillin	0.75 (0.75–1.00)	12 (71)	0 (0)	12 (71)
A07AA11–Rifaximin	1.17 (1.00–1.33)	2 (17)	6 (50)	8 (67)
J01AA02–Doxycycline	2.00 (1.00–2.00)	0 (0)	3 (60)	3 (60)
J01MA12–Levofloxacin	1.00 (1.00–1.50)	0 (0)	29 (45)	29 (45)
J01XX01–Fosfomycin	1.00 (0.50–1.00)	5 (45)	0 (0)	5 (45)
J01DD14–Ceftibuten	1.00 (1.00–1.00)	1 (20)	0 (0)	1 (20)
J01DD08–Cefixime	1.00 (1.00–1.00)	1 (6)	1 (6)	2 (12)
J01MA02–Ciprofloxacin	1.00 (1.00–1.00)	2 (2)	8 (9)	10 (11)
J01MA14–Moxifloxacin	1.00 (1.00–1.00)	0 (0)	0 (0)	0 (0)
J01MA01–Ofloxacin	1.00 (1.00–1.00)	0 (0)	0 (0)	0 (0)
J01MA06–Norfloxacin	1.00 (1.00–1.00)	0 (0)	0 (0)	0 (0)
J01EE01–Sulfamethoxazole plus trimethoprim	1.00 (1.00–1.00)	0 (0)	0 (0)	0 (0)

**Table 4 antibiotics-09-00038-t004:** Sub-use (PDD/DDD < 1.0) and overuse (PDD/DDD > 1.0) by pharmacological group. Data are described as n (%).

Pharmacological Group	N	Sub-Use	Overuse	Total
Macrolides	71	4 (6)	63 (89)	67 (95)
Lincosamides	60	55 (92)	0 (0)	55 (92)
Aminoglycosides	9	8 (89)	0 (0)	8 (89)
Nitrofurans	16	3 (19)	10 (63)	13 (82)
Cephalosporins	119	72 (61)	15 (13)	87 (74)
Tetracyclines	10	4 (40)	3 (30)	7 (70)
Rifamycins	12	2 (17)	6 (50)	8 (67)
Penicillins	160	28 (18)	119 (47)	147 (65)
Phosphonates	11	5 (45)	0 (0)	5 (45)
Quinolones	190	9 (5)	37 (19)	46 (24)
Sulfonamides	26	5 (19)	3 (12)	8 (31)
Carbapenems	1	0 (0)	0 (0)	0 (0)

**Table 5 antibiotics-09-00038-t005:** Normalized PDD with respect to DDD for antibiotics with at least five prescriptions. Data are reported as median (percentile 25–percentile 75).

Group	ATC Class–Name	(PDD-DDD)/DDD	p
Quinolones	J01MB02–Nalidixic acid	–0.63 (–0.63 to –0.63)	0.014
	J01MA12–Levofloxacin	0.00 (0.00–0.50)	<0.001
	J01MA02–Ciprofloxacin	0.00 (0.00–0.00)	0.052
	J01MA14–Moxifloxacin	0.00 (0.00–0.00)	1.000
	J01MA01–Ofloxacin	0.00 (0.00–0.00)	1.000
	J01MA06–Norfloxacin	0.00 (0.00–0.00)	1.000
Penicillins	J01CR02–Amoxicillin plus clavulanic acid	0.75 (0.75–0.75)	<0.001
	J01CA04–Amoxicillin	0.50 (0.50–0.50)	<0.001
	J01CA01–Ampicillin	–0.25 (–0.25–0.00)	0.182
	J01CF01–Dicloxacillin	–0.25 (–0.25–0.00)	0.182
Cephalosporins	J01DC02–Cefuroxime	1.00 (1.00–1.00)	0.003
	J01DD04–Ceftriaxone	–0.50 (–0.50 to –0.50)	<0.001
	J01DB01–Cefalexin	–0.25 (–0.25 to –0.25)	<0.001
	J01DD08–Cefixime	0.00 (0.00–0.00)	0.655
	J01DD14–Ceftibuten	0.00 (0.00–0.00)	0.317
Macrolides	J01FA09–Clarithromycin	1.00 (1.00–1.00)	<0.001
	J01FA10–Azithromycin	0.67 (0.67–0.67)	<0.001
	J01FA01–Erythromycin	0.50 (0.50–1.00)	0.168
	J01FA02–Spiramycin	–0.25 (–0.25–0.00)	0.577
Lincosamides	J01FF02–Lincomycin	–0.67 (–0.67 to –0.67)	0.003
	J01FF01–Clindamycin	–0.25 (–0.25 to –0.25)	<0.001
Sulfonamides	J01EE01–Sulfamethoxazole plus trimethoprim	0.00 (0.00–0.00)	0.942
Nitrofurans	J01XE01–Nitrofurantoin	0.50 (0.00–1.00)	0.064
Rifamycins	A07AA11–Rifaximin	0.17 (0.00–0.33)	0.107
Phosphonates	J01XX01–Fosfomycin	0.00 (–0.50–0.00)	0.038
Aminoglycosides	J01GB03–Gentamicin	–0.33 (–0.33 to –0.33)	0.020
Tetracyclines	J01AA02–Doxycycline	1.00 (0.00–1.00)	0.083
Carbapenems	J01DH03–Ertapenem	0.00 (0.00–0.00)	1.000

ATC = anatomical, therapeutical, and chemical.

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
