# Peer review of "Prevalence of Antibiotics Prescription and Assessment of Prescribed Daily Dose in Outpatients from Mexico City"

_antibiotics, 2020, doi:10.3390/antibiotics9010038_

Round 1

Reviewer 1 Report

Sánchez-Huesca et. al. analyzes the prevalence of antibiotics prescription as overuse of antibiotics is directly linked to antibiotic resistance. For this they used the data from outpatient patients who are prescribed the antibiotics. They also conclude that the prevalence of antibiotics subscription is significantly higher than the WHO recommendation’s and both over use and sub-use are higher. Study is also important as it tracks the prescription for outpatient but not the consumption outside prescription.

Overall study is meticulous and clearly presented author can take care of some of the minor points to enhance the highlight of their finding.

Minor points:

Table describing number of prescriptions with over and sub use for each class can be sorted based on total number of sub-use and over-use so it is easier to follow and focus on top candidates. Author may consider to merge “study limitation” and “perspective” within discussion as in current setting it takes away the well written message of discussion. Perspective can be in the last part of discussion to summarize the value of current work with emphasize for future directions within the discussion.

Author Response

Comment 1: Sánchez-Huesca et. al. analyzes the prevalence of antibiotics prescription as overuse of antibiotics is directly linked to antibiotic resistance. For this they used the data from outpatient patients who are prescribed the antibiotics. They also conclude that the prevalence of antibiotics subscription is significantly higher than the WHO recommendation’s and both over use and sub-use are higher. Study is also important as it tracks the prescription for outpatient but not the consumption outside prescription. Overall study is meticulous and clearly presented author can take care of some of the minor points to enhance the highlight of their finding.

Response: We appreciate the reviewer’s objective and constructive comments about our work. Below there is a response to the points raised by the reviewer.

Comment 2: Minor points: Table describing number of prescriptions with over and sub use for each class can be sorted based on total number of sub-use and over-use so it is easier to follow and focus on top candidates. 

Response: The results in Table 3 and Table 4 of the revised manuscript are sorted based on the total percentage of sub-use and over-use.

Comment 3: Author may consider to merge “study limitation” and “perspective” within discussion as in current setting it takes away the well written message of discussion. Perspective can be in the last part of discussion to summarize the value of current work with emphasize for future directions within the discussion.

Response: We eliminated the headers of sub-sections “3.1 Study limitations” and “3.2 Perspectives” to merge the study limitations and perspectives within the Discussion section. As the reviewer suggested, the last two paragraphs of the Discussion section emphasize the study perspectives.

Reviewer 2 Report

The study “Prevalence of Antibiotics Prescription and Assessment of Prescribed Daily Dose in Outpatients from Mexico City” estimates the prevalence of antibiotic usage by outpatients from Mexico City and reports a current discrepancy between two pharmacoepidemiological indicators: prescribed daily dose (PDD) and defined daily dose (DDD) as reported by the WHO.

Though the study is simple and does not gather much information (has some limitations, as stated by the authors – there is no information on the condition/type of infection for which the antibiotic was prescribed for), it may still have interest to be published as it gives a raw overview of antibiotic prescription scenario (pharmacological groups, sub-use or overuse) in Mexico City, and that general information is needed to build the basis for creating actions/measures to improve antibiotic prescription.

Specific and minor comments:

Table 1: It should be stated the pharmacological group of each antibiotic listed in the group of “other pharmacological groups”. Since it is referred that 12 pharmacological groups were included in this study (Line 60), it would be of interest having all groups named in the Table.  Moreover, in Tables 4 and 5 only 11 pharmacological groups are mentioned… Please explain. Lines 93-94: Correct as follows: “…prescriptions followed by macrolides, whose variations represent 94% as sub-use or overuse.” Lines 94-95: Correct as follows: “Moreover, huge discrepancy rates were presented regarding both lincosamides and quinolones when comparing PDD and DDD values.” Lines 101-105: “The antibiotics that showed the greatest discrepancy in the dose prescription were amoxicillin whether it is alone or in combination with clavulanic acid, levofloxacin, cefalexin, ceftriaxone, azithromycin, clarithromycin and clindamycin and with a lower rate some other drugs such as nalidixic acid, dicloxacillin, cefuroxime, lincomycin, fosfomycin and gentamicin.” The antibiotics listed as those having the greatest or the lowest discrepancy does not match with my interpretation taking into account the values presented in Table 5… Please check that. Lines 150-152: “For example, there is evidence that has already exposed the association of a high antibiotic prescription rate, low prescriber training, short time of consultation and prescription from a general practitioner”. This sentence is not very clear. Please rephrase for clarity.

Author Response

Comment 1: The study “Prevalence of Antibiotics Prescription and Assessment of Prescribed Daily Dose in Outpatients from Mexico City” estimates the prevalence of antibiotic usage by outpatients from Mexico City and reports a current discrepancy between two pharmacoepidemiological indicators: prescribed daily dose (PDD) and defined daily dose (DDD) as reported by the WHO.

Though the study is simple and does not gather much information (has some limitations, as stated by the authors – there is no information on the condition/type of infection for which the antibiotic was prescribed for), it may still have interest to be published as it gives a raw overview of antibiotic prescription scenario (pharmacological groups, sub-use or overuse) in Mexico City, and that general information is needed to build the basis for creating actions/measures to improve antibiotic prescription.

Response: We appreciate the reviewer’s objective and constructive comments, which helped us to improve the presentation of our work. Below there is a response to his/her recommendations.

Comment 2: Specific and minor comments: Table 1: It should be stated the pharmacological group of each antibiotic listed in the group of “other pharmacological groups”. Since it is referred that 12 pharmacological groups were included in this study (Line 60), it would be of interest having all groups named in the Table. Moreover, in Tables 4 and 5 only 11 pharmacological groups are mentioned… Please explain.   

Response: The results in Table 1 are now listed according to each pharmacological group, regarding the 12 pharmacological groups included in the study. The results of carbapenems were added to Tables 4 and 5 to include the 12 pharmacological groups.

Comment 3: Lines 93-94: Correct as follows: “…prescriptions followed by macrolides, whose variations represent 94% as sub-use or overuse.” Lines 94-95: Correct as follows: “Moreover, huge discrepancy rates were presented regarding both lincosamides and quinolones when comparing PDD and DDD values.”

Response: The text was corrected, as indicated by the reviewer.

Comment 4: Lines 101-105: “The antibiotics that showed the greatest discrepancy in the dose prescription were amoxicillin whether it is alone or in combination with clavulanic acid, levofloxacin, cefalexin, ceftriaxone, azithromycin, clarithromycin and clindamycin and with a lower rate some other drugs such as nalidixic acid, dicloxacillin, cefuroxime, lincomycin, fosfomycin and gentamicin.” The antibiotics listed as those having the greatest or the lowest discrepancy does not match with my interpretation taking into account the values presented in Table 5… Please check that.

Response: Thank you very much for addressing this point. We updated the description of Table 5 with a correct description of the antibiotics with significant discrepancies (lines 102 to 107). The ATC class – names in Table 5 were re-organized within each pharmacological group to show first the antibiotics with significant discrepancies (p < 0.05) at the top of each list and ordered by discrepancy magnitude.

Comment 5: Lines 150-152: “For example, there is evidence that has already exposed the association of a high antibiotic prescription rate, low prescriber training, short time of consultation and prescription from a general practitioner”. This sentence is not very clear. Please rephrase for clarity.

Response: The sentence was rephrased for clarity as follows: “For example, among general practitioners, several factors are associated with the antibiotic prescribing volume, including appointment duration, training practice, as well as the prescriber’s age and sex” (lines 152 to 154).

Reviewer 3 Report

The analysis presented by Sanchez-Huesca et al. is interesting and addresses a very actual topic. Drug resistances are spreading rapidly and one of the causes is the misuse of antibiotics. The absence of an electronic record of prescriptions is an important obstacle to supervision, as pointed out by authors. So, the huge work done in collecting all those data deserves consideration and hope to be a suggestion for further improvements for the National Health System of Mexico City.

The manuscript is written clearly, but results are not very immediate. I suggest adding or replacing some tables (i.e. Table 3) with graphs.

Author Response

Comment 1: The analysis presented by Sanchez-Huesca et al. is interesting and addresses a very actual topic. Drug resistances are spreading rapidly and one of the causes is the misuse of antibiotics. The absence of an electronic record of prescriptions is an important obstacle to supervision, as pointed out by authors. So, the huge work done in collecting all those data deserves consideration and hope to be a suggestion for further improvements for the National Health System of Mexico City.

Response: We appreciate the reviewer’s objective and constructive comments about our work.

Comment 2: The manuscript is written clearly, but results are not very immediate. I suggest adding or replacing some tables (i.e. Table 3) with graphs.

Response: After considering several possible graphical representations of the results, we realized that the graphs would be too complex (particularly for Table 3, which comprises two different variables, the ratio PDD/DDD and the percentage of sub-use and over-use). Several changes were made in the presentation of results to improve clarity. The results in Table 3 and Table 4 are now sorted based on the total percentage of sub-use and over-use. Also, we updated the description of Table 5 with a correct description of the antibiotics with significant discrepancies (lines 102 to 107). The ATC class – names in Table 5 were re-organized within each pharmacological group to show first the antibiotics with significant discrepancies (p < 0.05) at the top of each list and ordered by discrepancy magnitude. We hope that results became more immediate after these changes.